# A decadal, hourly high-resolution satellite dataset of aerosol optical properties over East Asia

Jeewoo Lee<sup>1</sup>, Jhoon Kim<sup>1</sup>, Seoyoung Lee<sup>2,3</sup>, Myungje Choi<sup>2,3</sup>, Jaehwa Lee<sup>3,4</sup>, Daniel J. Jacob<sup>5</sup>, Su Keun Kuk<sup>6</sup>, Young-Je Park<sup>7</sup>

- <sup>5</sup> Department of Atmospheric Sciences, Yonsei University, Seoul, Republic of Korea
  - <sup>2</sup>Goddard Earth Sciences Technology and Research II, University of Maryland Baltimore County, Baltimore, MD, United States
  - <sup>3</sup>NASA Goddard Space Flight Center, Greenbelt, MD, United States
  - <sup>4</sup>Earth System Science Interdisciplinary Center, University of Maryland, College Park, MD, United States
- 10 <sup>5</sup>School of Engineering and Applied Sciences, Harvard University, Cambridge, United States
  - <sup>6</sup>Samsung Particulate Matter Research Institute, Samsung Advanced Institute of Technology, Gyeonggi-do, Republic of Korea <sup>7</sup>TelePIX Co., Ltd., Seoul, Republic of Korea

Correspondence to: Jhoon Kim (jkim2@yonsei.ac.kr)

**Abstract.** Formerly known as one of the most polluted regions of the globe, East Asia underwent a dramatic improvement of air quality, especially for aerosols, starting in the 2010s. Numerous satellites have observed East Asia for a long time duration, but often with a low spatial or temporal resolution, limiting their ability to capture small-scale variabilities or provide continuous observations of long-range transport of aerosols. In this study, we provide an hourly aerosol optical property (AOP) dataset retrieved from the Korean Geostationary Ocean Color Imager (GOCI), with a high spatial resolution of 2 km at nadir, covering the entire operational period from March 2011-March 2021. The dataset is retrieved using the Yonsei Aerosol Retrieval Algorithm, providing aerosol optical depth (AOD) at 550 nm as the primary product, along with fine mode fraction, single scattering albedo, Ångström exponent, and aerosol type as ancillary products. Seasonal validation of AOD against the Aerosol Robotic Network (AERONET) showed that the fraction of data points within the expected error range of 0.05 + 15% varied from 56.4% in June-July-August to 64.5% in September-October-December, with the mean bias generally within ±0.05. Compared to the operational version, the high-resolution product demonstrated improved retrieval capability in the presence of broken clouds, along complex coastlines, and in capturing AOD variability at the sub-district level. The decadal AOD exhibited a decreasing trend over four major cities within the observation domain. We expect this data to be widely used in climate modelling, reanalysis, atmospheric chemistry, marine optics, environmental health studies, variability and trend analysis, contributing to a more comprehensive understanding of the interactions between climate change, trace gases, human health, and AOPs. The dataset presented in this work is publicly available for download https://doi.org/10.5281/zenodo.16656274 (Lee et al., 2025).

## 1 Introduction

55

Atmospheric aerosols play an important role in air pollution and climate change. The presence of ambient aerosols reduces visibility, and the exposure deteriorates human health by increasing the mortality rate and the probability of cardiovascular diseases, asthma, and neurodegenerative diseases (Charlson, 1969; Dockery et al., 1992; Baldacci et al., 2015; Wang et al., 2017; Pye et al., 2021). In addition, aerosols can act as a medium for atmospheric chemical reactions, influencing the concentration of atmospheric trace gases (Andreae and Crutzen, 1997). Importantly, aerosols are one of the major components that offset the warming effect of greenhouse gases by scattering solar radiation, exerting a net cooling effect (Kaufman et al., 2002; Bellouin et al., 2020; IPCC, 2021). From 1750 to 2019, the effective radiative forcing of atmospheric aerosols in total is estimated to be -1.1 W m<sup>-2</sup>, compared to +2.16 W m<sup>-2</sup> for carbon dioxide (IPCC, 2021). Thus, long-term data of aerosol optical properties (AOPs) can serve as a fundamental resource for research across various fields.

There have been extensive studies on the effects and distribution of aerosols especially over East Asia, as the region is renowned for massive emissions of aerosols and their precursors. To this end, air quality campaigns such as the Distributed Regional Aerosol Gridded Observation Networks - Northeast (DRAGON-NE) Asia 2012 campaign and the Korea-United States Air Quality (KORUS-AQ) field study that leveraged airplanes and surface sites equipped with various in-situ and remote sensing measurement instruments were conducted to investigate the atmospheric environment over Northeast Asia (Holben et al., 2018; Crawford et al., 2021). According to the World Health Organization (WHO), South and East Asia are the most polluted regions in terms of surface-level particulate matter (PM), showing the highest annual population-weighted particulate matter smaller than 2.5 µm in diameter (PM<sub>2.5</sub>) in 2019 (WHO, 2021). This poses a major threat over East Asia, compared to Europe, North America, and the Western Pacific, where the exposure to particulates is decreasing. While approximately 0.2 million excess deaths were estimated in the United States and Canada due to aerosol exposure, China, India, and other parts of Asia were expected to experience 2.4 million, 2.2 million, 1.3 million excess deaths, respectively, due to high aerosol concentration (Burnett et al., 2018). As a result, East Asian countries have implemented policies aimed at managing air quality. For instance, China implemented the Clean Air Action in 2013, which resulted in a steep reduction of anthropogenic air pollutants, including aerosols and their precursors (Zheng et al., 2018). Similarly, South Korea has consistently adopted air quality improvement plans for its capital city, the Seoul Metropolitan Area (SMA), since 2005 (Korea Ministry of Environment or KMOE, 2005; KMOE, 2013). SMA achieved an 8% and 10% reduction in mortality rates from cardiovascular and cerebrovascular diseases through PM reduction in 2012-2013 compared to 2004-2005 (Han et al., 2018). Air quality concerns are not confined to individual countries but rather represent a cross-boundary challenge. In addition, strong westerlies over Northeastern Asia facilitate long-range transport of aerosols, adding complexity to studies on aerosol distribution (Lee et al., 2019; Lee et al., 2021).

Satellite observations provide spatially continuous data unlike in-situ measurements. Primarily, they provide the amount of atmospheric aerosols as aerosol optical depth (AOD), a measure of aerosol extinction within a vertical column of air from the Earth's surface to the top-of atmosphere (TOA). For example, the Moderate Resolution Imaging Spectroradiometer (MODIS)

onboard Terra and Aqua views the sunlit side of the entire globe every one or two days. MODIS AOD products, such as Dark Target (DT), Deep Blue (DB), and Multi-Angle Implementation of Atmospheric Correction (MAIAC), have demonstrated strong agreement with ground-based observations such as the Aerosol Robotic Network (AERONET) (Levy et al., 2010; Hsu et al., 2013; Levy et al., 2013; Lyapustin et al., 2018; Hsu et al., 2019). To continue the long-term AOD records, these algorithms have been implemented on other satellites, such as the Visible Infrared Imaging Radiometer Suite (VIIRS) (Hsu et al., 2019; Sawyer et al., 2020; Lee et al., 2024) with cross-calibration to MODIS (Lyapustin et al., 2023). However, because of the coarse temporal resolution, these low Earth orbit (LEO) satellites are less effective for studying diurnal variations and instantaneous peaks of observables (International Ocean-Colour Coordinating Group (IOCCG), 2012; Kim et al., 2020). In contrast, Geostationary Earth Orbit (GEO) satellites observe a certain domain with a high spatial and temporal resolution. Many GEO satellites such as Himawari-8/9, the Meteosat series, and the Geostationary Operational Environmental Satellites have observed their respective field of regard (FOR) to provide diurnal information of atmospheric aerosols. These long-term diurnal observations are especially valuable for regions such as East Asia, where the aerosol distribution and concentration are a major concern. For instance, the Geostationary Ocean Color Imager (GOCI) onboard the Communication, Ocean and Meteorological Satellite (COMS, also known as GK-1), the first ocean color imager in GEO was launched in 2010 and observed East Asia for 10 years. It observed East Asia eight times hourly during daytime, from 00 UTC to 07 UTC, with a spatial resolution of 500 m × 500 m at nadir. GOCI was equipped with six visible and two near-infrared (NIR) bands, ranging from 412 nm to 865 nm. This multispectral information provides advantages for aerosol retrieval compared to the conventional 5-channel Meteorological Imager (MI) on the same platform with only one visible channel, offering relatively limited information content for aerosol retrieval (Kim et al., 2016). The operational algorithm for GOCI aerosol retrieval, the Yonsei Aerosol Retrieval Algorithm (YAER), provided hourly AOD over cloud-free and snow/ice-free pixels with a 6 km × 6 km

The GOCI YAER product has been widely used in aerosol studies, contributing not only to regional analyses over East Asia but also to a deeper understanding of aerosols in general. For instance, surface-level PM<sub>2.5</sub> concentration was estimated through the integration of chemical transport models or data assimilation in East Asia (Saide et al., 2014; Xu et al., 2015; Pang et al., 2018; Pendergrass et al., 2025). The products also played a crucial role in unravelling the aerosol properties during the DRAGON-NE Asia and the KORUS-AQ campaigns (Choi et al., 2016; Lennartson et al., 2018; Choi et al., 2019; Lim et al., 2021).

resolution for data quality assurance (Lee et al., 2010a; Choi et al., 2016; Choi et al., 2018).

To continue the GOCI mission, GOCI-II was launched in February 2020. Driven by improved sensor technology, GOCI-II now features a higher spatial resolution of 250 m × 250 m at nadir, effectively meeting the evolving needs of present-day users. According to Lee et al. (2023), the higher resolution of the GOCI-II YAER product has increased the amount of AOD data by a factor of five compared to the GOCI YAER operational product, enabling more detailed atmospheric monitoring. Building upon these technological advancements, some studies have pointed out the need for higher spatiotemporal resolutions of satellite products for air quality assessments (Lee et al., 2024). Other GEO missions such as the Advanced Meteorological Imager and the Geostationary Environment Monitoring Spectrometer were also launched in the same era of GOCI-II to monitor

the atmosphere of East Asia (Kim et al., 2020; Kim et al., 2024). Over these state-of-the-art satellites, GOCI has a relative advantage on its diverse spectral bands in the visible spectrum, its long-term observation records with stability, and the presence of a successor satellite to continue its mission. The presence of multiple visible bands allows GOCI to efficiently separate the signal between the atmosphere and the Earth surface. In addition, long-term records of aerosols from a same algorithm framework can be obtained when GOCI and GOCI-II observations are combined. Furthermore, it meets the sustained demand for high-resolution AOP datasets over the long term in air quality research and related applications, including aerosol-cloud interaction and epidemiological studies (e.g. Eck et al., 2020; Pu and Yoo, 2022).

Here, we present a high-resolution, decadal hourly AOPs dataset over East Asia based on GOCI YAER, which is challenging with respect to data quality due to limited number of pixels available. The spatial resolution of the product has been increased ninefold (2 km × 2 km) compared to the operational version (6 km × 6 km). Validation against AERONET, analysis of diurnal variation, and cases studies highlighting the strengths of high-resolution data are provided. The dataset and high-resolution YAER algorithm are described in Section 2. Section 3 presents the characteristics of the AOPs, validation results and analysis of the data record. The availability of data is presented in Section 4. Section 5 summarizes and concludes the study.

# 2 Materials and methodology

## 2.1 Retrieval algorithm

100

115

The GOCI high-resolution YAER goes through three main steps to retrieve the hourly AOPs dataset. The inputs used for retrieval are listed in Table 1.

Table 1. List of input data for GOCI high-resolution YAER.

| Input data                                             | Usage                                                |
|--------------------------------------------------------|------------------------------------------------------|
| GOCI Level 1B                                          | Step 1: TOA reflectance calculation                  |
| ECMWF ERA 5 wind speed climatology                     | Step 3: inversion over ocean                         |
| Land surface reflectance climatology                   | Step 3: inversion over land and turbid water         |
| Global Multi-resolution Terrain Elevation Data (GMTED) | Step 3: inversion over land, ocean, and turbid water |
| Ocean LUT                                              | Step 3: inversion over ocean                         |
| Land LUT                                               | Step 3: inversion over land and turbid water         |

#### 2.1.1 Step 1: Cloud detection and pixel aggregation

The cloud detection for the high-resolution GOCI dataset is the same as used for the operational dataset, as described in Choi et al. (2018). A brief description is provided here. The algorithm reads hourly GOCI observed radiances of wavelengths

centered at 412, 443, 490, 555, 660, 680, 745, and 865 nm along with geolocation and observation geometry information. These values have a spatial resolution of 500 m × 500 m at nadir ('observation resolution'). Then, the TOA reflectance of each wavelength can be calculated using eq. (1):

$$\rho_{\lambda} = \frac{\pi \cdot L_{\lambda}}{\mu_0 \cdot E_{\lambda}},\tag{1}$$

where  $\lambda$  is the GOCI wavelength,  $\mu_{\theta}$  is the cosine of solar zenith angle,  $L_{\lambda}$  is the observed radiance from GOCI, and  $E_{\lambda}$  is the solar extraterrestrial flux.

In this step, the spectral TOA reflectance is used to detect and remove pixels that are inappropriate for aerosol retrieval, such as clouds, snow, and bright surfaces, which cannot be distinguished from aerosol signals in most situations in the visible spectra. Clouds are bright in visible wavelengths, and cloud edges and broken clouds are spatially inhomogeneous compared to clear-sky pixels. Considering these physical and optical properties of clouds, clouds over land and ocean are detected separately. For clouds over ocean, a TOA reflectance threshold test and a 3 × 3 pixel standard deviation test are applied. For clouds over land, the detection scheme can be classified into three categories, which are TOA reflectance threshold test, reflectance ratio test, and the 3 × 3 pixel standard deviation and mean test. The tests are conducted over multiple wavelengths and ratios (Choi et al., 2018). The thresholds are determined based on the references of each cloud detection scheme considering its compatibility to GOCI. Land pixels with NDVI less than -0.01 are regarded as inland water and thus are removed. In addition, pixels affected by sun glints over water are removed. After the cloud detection, dusty pixels and heavy aerosols initially misclassified as clouds are reinstated based on TOA reflectance ratios and 3 × 3 pixel standard deviation and mean.

The remaining clear-sky pixels are converted into 2 km  $\times$  2 km resolution ('product resolution') for aerosol retrieval by aggregating 4  $\times$  4 observation pixels. When collecting the clear-sky pixels within the 4  $\times$  4 box, the brightest 60% and the darkest 20% of pixels within the box are excluded from averaging to minimize residual cloud effects or cloud shadows. Additional cloud detection and bright surface removal is conducted at the product resolution. The cloud detection thresholds and criteria of the high-resolution algorithm aligns with that of the operational counterpart.

It should be noted that the product resolution was set by weighing the benefits of high spatial resolution against its potential drawbacks. High product resolution may result in a decrease in signal-to-noise ratio, an increase in cloud 3D effects, and heightened sensitivity to radiometric noise. Furthermore, the associated increase in retrieval uncertainty often undermines its benefit, offering limited added value in the context of the intended applications. Due to these reasons, we selected 2 km as the product resolution.

Geospatial variables needed for retrieval, such as longitude, latitude, land-sea mask, geometry, are averaged and aggregated into the product resolution without screening.

#### 150 2.1.2 Step 2: Classification of surface regimes

To allocate the clear-sky product resolution pixels into their corresponding surface regimes, the land-sea mask is used to discriminate land and ocean pixels. The ocean pixels are further divided into turbid water and the open ocean (hereafter ocean)

using the  $\Delta\rho_{660}$ . The  $\Delta\rho_{660}$  is the difference between TOA reflectance at 660 nm and the interpolated value between 412 nm and 865 nm at 660 nm (Li et al., 2003; Choi et al., 2016). Ocean pixels with  $\Delta\rho_{660}$  between -0.05 and -0.01, and high TOA reflectance at 660 nm are classified as turbid water and are put into the turbid water algorithm.

For both the land and turbid water algorithms, the surface reflectance is calculated based on the minimum reflectance technique (Koelemeijer et al., 2003; Hsu et al., 2004). We calculated the Rayleigh-corrected reflectance (RCR) for every GOCI observations at the observation resolution ranging from March 2011 to February 2016. The minimum reflectance technique assumes that there would be at least a single cloud-free, aerosol-free scene within the sampling period, which can be represented by the darkest values of RCR. While the 5-year period provides plenty of observations to find a clear scene, it should be noted that no day can truly be aerosol-free, and the least amount of aerosol present in the 'clear scene' may cause a marginal overestimation of surface reflectance. Here, we averaged the darkest 1–3% of RCR for each month and wavelength as the *a priori* of GOCI monthly surface reflectance.

The ocean algorithm adopts the Cox and Munk method, where the wind speed of the water surface is used to calculate the surface reflectance (Cox and Munk, 1954). We collected the wind speed at 10 m above sea level from the European Centre for Medium-Range Weather Forecasts Interim dataset. The dataset has  $0.25^{\circ} \times 0.25^{\circ}$  resolution, and a 5-year average of each month was taken as the *a priori*. For both land and ocean, the *a priori* are set to the value for the 15<sup>th</sup> day of each month, and the values for other days are calculated by linearly interpolating the *a priori* of two adjacent months.

# 2.1.3 Step 3: Inversion

- The GOCI high-resolution YAER utilizes the LUT to efficiently retrieve AOD from vast number of pixels. Two LUTs, the land LUT and ocean LUT, are constructed using libRadtran radiative transfer package (Mayer and Kylling, 2005). The land LUT is composed of wavelength, solar zenith angle (SZA), satellite viewing zenith angle (VZA), relative azimuth angle (RAA), AOD, surface reflectance, surface altitude, and aerosol type nodes, and that of ocean is composed of wavelength, SZA, VZA, RAA, AOD, wind speed, and aerosol type. Here, the LUT is composed of pre-computed TOA reflectance from the radiative transfer model, calculated for each node. AOD is retrieved by interpolating the node variables that are the closest match of each retrieval pixel. The aerosol models are derived from AERONET AOP climatology prior to the GOCI observation period. The AOD nodes within the LUT range from 0 to 3.6, and negative AOD values due to extrapolation are permitted down to 0.05. 26 aerosol models were assumed, which are classified into three absorbing criteria and nine size criteria according to SSA and FMF, respectively.
- During the inversion process, different wavelengths are used for different surface regimes. For land, wavelengths 412 to 680 nm are used for inversion, and only pixels with surface reflectance lower than 0.15 at these selected wavelengths are used in the inversion process. The two NIR wavelengths, 745 and 865 nm, are excluded because of their high surface reflectance and the resulting uncertainties in separating the signals of the land surface and atmospheric aerosols. For ocean and turbid water, wavelengths 412, 443, 745, and 865 nm, and 412 and 865 nm are selected, respectively, to avoid channels with high water-leaving radiance (Ahn et al., 2012). For all surface regimes, only pixels with SZA smaller than 70° are used. The algorithm

selects three aerosol models that have the lowest standard deviation of 550 nm AODs retrieved from individual spectral bands. Then, the spectrally averaged 550 nm AODs for each of these three models are used to calculate the final AOD, which is determined as a weighted mean of the averaged AODs, with weights based on the inverse of the standard deviation. FMF, SSA, AE are determined based on the selected aerosol models and their corresponding weights. The aerosol type is then classified into six types from FMF and SSA thresholds: dust, non-absorbing coarse, mixture, highly-absorbing fine, moderately-absorbing fine, and non-absorbing fine (Lee et al., 2010b).

## 2.2 Dataset description

The GOCI high-resolution AOPs are provided for clear-sky pixels over land, ocean, and turbid water. The FOR of GOCI is shown in Fig. 1. The wedge pattern on the left and right edges of Fig 1 results from the 16 slots comprising the observation (Ryu et al., 2012). In the retrieval process and the resulting dataset, all slots are merged into a single scene for the user's convenience. The dataset ranges from March 1<sup>st</sup>, 00:30 UTC, 2011 to March 31<sup>st</sup>, 07:30 UTC, 2021. The proposed GOCI YAER high-resolution AOPs dataset consists of five properties: AOD at 550 nm, single scattering albedo (SSA) at 440 nm, fine mode fraction (FMF) at 550 nm, Ångström exponent (AE) between 440 nm and 870 nm, and aerosol type. Unless otherwise stated, the AOD, FMF, SSA, and AE of in this study are at the wavelength of GOCI YAER. The AOD is retrieved by comparing measured and calculated TOA reflectance, and the other four variables are derived from selected aerosol models for AOD retrieval. The SSA quantifies the measure of aerosol scattering properties, defined as the ratio of aerosol scattering to extinction. FMF and AE are the measure of aerosol size, which can act as a proxy for anthropogenic aerosols (Anderson et al., 2005). Aerosol type is determined from the SSA and FMF, following the work of Lee et al. (2010b). Since FMF, SSA, AE, and aerosol type are ancillary variables, it is recommended that these variables be used qualitatively or for interpreting the AOD error. Pixels that were not used in the retrieval are represented as NaN. The notation of the variables within the dataset and their respective dimensions are listed in Table 2.

Due to the large file size, the dataset is saved in zipped files in tar.gz format, on a monthly basis. The zipped files are named following the "GOCI\_YAERAERO\_hires\_{YYYYMOMO}.tar.gz" convention, where YYYY and MoMo are 4-digit numeric year and 2-digit month, respectively. The zipped files include hourly dataset in NetCDF-4 format, and the file names are structured following the "GOCI\_YAERAERO\_hires\_{YYYYMOMODDHHMiMiSS}.nc" convention, where DD, HH, MiMi, SS indicate the 2-digit numeric day, hour, minute, and seconds of GOCI observation time.

Table 2. List of variables in the GOCI high-resolution AOPs product.

| Variable name         | Long description                   | dimensions                          |
|-----------------------|------------------------------------|-------------------------------------|
| Aerosol_Optical_Depth | Aerosol Optical Depth at 550 nm    | 2D; (latitude=1421, longitude=1391) |
| Aerosol_Type          | Aerosol type: 1 = Dust, 2 = Non-   | 2D; (latitude=1421, longitude=1391) |
|                       | absorbing Coarse, 3 = Mixture, 4 = |                                     |

|                          | High-absorbing Fine, 5 = Moderate- |                                     |
|--------------------------|------------------------------------|-------------------------------------|
|                          | absorbing Fine, 6 = Non-absorbing  |                                     |
|                          | Fine                               |                                     |
| Angstrom_Exponent        | Calculated Angstrom Exponent       | 2D; (latitude=1421, longitude=1391) |
|                          | between 440 and 870 nm             |                                     |
| Fine_Mode_Fraction       | Fine Mode Fraction at 550 nm       | 2D; (latitude=1421, longitude=1391) |
| Single_Scattering_Albedo | Single Scattering Albedo at 440 nm | 2D; (latitude=1421, longitude=1391) |

Figure 1. The field of regard of GOCI. The background represents the RGB composite of surface reflectance database of April, 03 UTC. The blue, green, red, yellow boxes indicate the location of  $0.3^{\circ} \times 0.3^{\circ}$  grid encompassing the four major cities within the GOCI domain, Beijing, Pyongyang, Seoul, and Tokyo, respectively. The magenta box indicates the southwestern coast of the Korean Peninsula shown in Fig. 3.

#### 3 Results and discussion

#### 3.1 Data characteristics

The characteristics and strengths of GOCI high-resolution AOD are presented in this section. Figure 2 shows an example of GOCI high-resolution and operational AOD over two cloudy scenes in the Pacific Ocean. Amongst the complex cloud structures, the AOD is retrieved over cloud-free pixels that pass the cloud masking algorithm. Compared to the operational AOD products (Fig. 2b and e), the fine-scale AOD between cloud structures is more pronounced in the high-resolution AOD

(Fig. 2a and d). The larger pixel size of the operational product (Fig. 2b and e) may make the retrieved area appear more extensive. However, this implies that the observation resolution pixels are incorporated to represent an area that encompasses farther locations during the 6 km aggregation process. Over a horizontal plane, the number of high-resolution pixels within the domain was more than double that of the operational version (Fig. 2c and f). In addition, the 2 km pixels are in close proximity of cloud pixels, providing a more precise information on AOD over cloud-free areas which are surrounded by clouds.

Figure 2. GOCI AOD on January 27th, 2018, 02 UTC (top panel) and March 11st, 2018, 00 UTC (bottom panel) over the Pacific Ocean. GOCI high-resolution AOD (a, d), the operational AOD (b, e) and the comparison of two data over a horizontal plane (c, f) are shown. Red lines over the four maps (a, b, c, d) indicate the longitudinal pixels used for constructing (c) and (f).

Figure 3 shows two cases over the southwestern coast of the Korean Peninsula, which is characterized by a complex coastline with countless small islands and bays, and frequent occurrences of turbid water. Studies on GOCI ocean color algorithms such as atmospheric correction, algae blooms, chlorophyll-a, and turbidity have focused on these regions (Ahn et al., 2012; Choi et al., 2012; Choi et al., 2014; Kim et al., 2016; Lee et al., 2021). Here, AOD information is critical for atmospheric correction, which is fundamental for earning oceanic optical properties. Therefore, we investigated the potential of high-resolution AOD products over complex coastlines and their comparative advantage over operational products.

Two characteristics stand out over the domain. First, the operational version of AOD of Fig. 3 shows a discrete, artifactual boundary over pixels with mixed terrain of island and open ocean. This is because a single coarse 6 km pixel over coastal areas reflects the combined signals from both island and ocean simultaneously. On the other hand, the high-resolution AOD pixels over small islands and the surrounding ocean are distinctly delineated, providing reliable information for both regimes. Second, the extent of turbid water pixels appearing as high AOD has decreased, due to a finer scale of land-sea mask resolution and

Figure 3. GOCI high-resolution AOD (left panel) and operational AOD (right panel) over the southwestern coast of the Korean Peninsula (magenta box, Fig. 1) for (a) May 14th, 2018, and (b) October 28th, 2018, respectively. Each set of three maps show the AOD of 00–02 UTC (top to bottom). Pink lines indicate the coastlines.

The competence of the high-resolution product is further examined for a highly populated megacity, namely the SMA of South Korea. Approximately half of the country's population reside in SMA, setting a large number of residents vulnerable to local air pollution (https://www.index.go.kr). Figure 4 shows an incident of medium- to high AOD case on May 9th, 2012, amid the

DRAGON-NE campaign. As extra AERONET sites were set up during the campaign, maximum of 10 sites including the sites with long-term records are shown. The unique feature of the high-resolution product is the masked pixels across the figures, which are due to the presence of the Han River. The respective 6 km pixels including the Han River were not screened out because although the inland water pixels in the observation resolution were masked out successfully, the adjacent land pixels within the same 6 km pixel met the aggregation criteria. In other words, some pixels that are included in the operational product but not retrieved in the high-resolution product may result from a mixture of valid and invalid observation resolution pixels. These mixed pixels may fail to meet the threshold for merging in the high-resolution retrieval but are included in the coarser operational resolution, which is merged from a greater number of pixels. As a result, certain pixels in the operational product may reflect signals from only a very small portion within the 6 km area. Therefore, it can be inferred that the 2 km products effectively eliminate the influence of inappropriate pixels in fine geographical features.

The AOD distributions of 2 km products (left panel of Fig. 4) and the 6 km products (right panel of Fig. 4) are similar, but more subtle features are depicted in the high-resolution products. The high-resolution pixels located within the corresponding 6 km pixel are not a simple super-resolution version of the 6 km version but effectively represents the sub-pixel variability within the 6 km pixel. Furthermore, while a single pixel of the 6 km product covers one or more sub-districts of SMA, multiple high-resolution 2 km pixels fall into a single sub-district, providing more specific information both socially and geographically.

Figure 4. GOCI high-resolution AOD (left two panels) and operational AOD (right two panels) for May 9<sup>th</sup>, 2012, over the SMA (37.4–37.7°N, 126.6–127.2°E). The red lines on the map indicate the sub-districts of SMA, and the triangles and the filled colors represents AERONET locations and their respective collocated AOD. The numbers located at upper left of each plot are observation times in UTC.

#### 3.2 Validation

We validated the GOCI high-resolution YAER AOD to AERONET AOD, which is widely employed as ground truth for satellite AOPs validation owing to its low uncertainties (Holben et al., 1998). In the visible spectra, the uncertainty of AERONET AOD is known to be ~0.010–0.021, which is much lower than that of satellites (Eck et al., 1999; Sinyuk et al., 2020). Here, we used the AERONET Version 3 Level 2.0 direct sun AOD for validation, and the AOD at 550 nm was obtained by a quadratic interpolation of spectral AERONET AOD (Giles et al., 2019; Sinyuk et al., 2020). For collocation of GOCI high-resolution YAER products and AERONET, AERONET observations within the ±30 minutes from GOCI observation time and the retrieved pixels within the 25 km radius around each AERONET site were averaged, respectively. AERONET sites for short-term temporary usage were excluded, leaving 39 sites for analysis. The expected error (EE) envelope is defined as EE =  $\pm$  (0.15 × AERONET AOD + 0.05), following the MODIS DT standards of Levy et al. (2013). Figure 5 shows the validation results of the whole GOCI observation period, divided into four seasons. For all four seasons, 56-66% of the collocated retrievals are within the EE envelope, indicating a reasonable accuracy compared to AERONET. Collocated points were most abundant during spring (Fig. 5b, March-May, MAM), which can be attributed to two causes; lower SZA and a reduced portion of clouds and snowy surfaces leading to an increase in retrieved GOCI pixels; additional AERONET sites were set up to support the DRAGON-NE campaign in spring 2012. The portion of GOCI and AERONET AOD >1.5 was 0.02%, 0.93%, 2.8%, and 0.85% for winter, spring, summer, and autumn, respectively. The higher portion of high AOD in summer is due to aerosol hygroscopic growth, driven by the high relative humidity over East Asia during this season (Zhai et al., 2021). Some points where low AERONET AOD and high GOCI AOD coincide are due to residual clouds, where they are misinterpreted as high AOD. This is more pronounced in summer, where clouds are dominant in Northeast Asia due to monsoon (Fig. 5c). Validation results divided into land and ocean are shown in Fig. A1 and A2, respectively. The validation criteria are the same for Fig. 5, but ocean and land pixels were masked when validating for land and ocean, respectively. Validation for ocean had better results, because there are fewer uncertainties arising from surface reflectance over open ocean. Figure 6 shows the fraction of GOCI AOD retrievals within the expected error and the Pearson correlation coefficient compared to AERONET measurements at individual sites. Although there were some variations, most sites showed a R > 0.8, % within EE > 60% and MBE 

Figure 5. Validation of GOCI high-resolution YAER AOD to AERONET AOD during the whole observation period, classified into four seasons: (a) winter, (b) spring, (c) summer, and (d) autumn. The number of collocated points (N), correlation coefficients (R), the root mean squared error (RMSE), and mean bias error (MBE) are shown for each plot.

Figure 6. Validation metrics for AERONET sites within the GOCI domain, for (a) the % within EE envelope, (b) R, (c) MBE, (d) mean absolute error (MAE), (e) RMSE, (f) N.

Figure 7 presents the bias of GOCI high-resolution YAER AOD compared to AERONET AOD. The points of Fig. 7a-c are the median of collocated records sorted in ascending order based on the x axis variable, divided into 19, 18, and 9 intervals for all domains, land, and ocean, respectively. For Fig. 7d, the median values of collocated records for each GOCI observation hour are shown. Land and ocean points are collected by masking out retrieved ocean and land pixels using the land-sea mask in the collocation process, respectively. For all four variables, the bias of GOCI AOD is mostly around -0.05–0.05 range, showing a good agreement to AERONET. When AERONET AOD is larger than 1, GOCI AOD slightly underestimates, with a bias of approximately -0.1 (Fig. 7a). The underestimation of high AOD is primarily attributed to errors in aerosol model selection and the errors of assumed aerosol optical properties within the LUT. Underestimation occurs especially when absorbing aerosols are mistaken as scattering aerosols, such as dust or non-absorbing coarse types. There were some overestimations in GOCI ocean retrievals in the NDVI range of -0.2—0.1, which is due to AOD overestimation over turbid water. Over land, some overestimation of AOD is observed presented over a low NDVI range (Fig. 7b). The overestimation of AOD over turbid water and sparsely vegetated land indicates that the surface reflectance *a priori* over these regions is underestimated. This indicates that when the underlying surface conditions are bright, the overestimation of AOD should be considered. The diurnal bias of GOCI AOD is minimal, where the lowest and highest bias are within the -0.05–0.05 range.

The underestimation of AOD during the morning and late afternoon can be attributed to extended light paths during these hours compared to other hours.

Figure 7. Bias of GOCI high-resolution YAER AOD compared to AERONET, according to (a) AERONET AOD, (b) NDVI, (c) scattering angle, and (d) GOCI observation time. The vertical bars of (d) represent the number of data collocated to AERONET.

The four ancillary variables, namely, FMF, SSA, AE, and aerosol type are evaluated quantitatively and qualitatively. First, the four variables were quantitatively validated to AERONET Version 3 Level 2.0 inversion products for the entire observation period, with additional AOD thresholds added to the spatial and temporal collocation criteria of AOD. AERONET data points with its quadratically interpolated 550 nm AOD larger than 0.3, 0.4, 0.3, 0.3 were used for validating FMF, SSA, AE, aerosol type, respectively, to ensure sufficient aerosol signals (Choi et al., 2018). The mode of GOCI aerosol type in the 25 km radius of AERONET site and the type determined from AERONET FMF and SSA are collocated for aerosol type comparison.

The validation results of ancillary variables are less promising compared to those of AOD at 550 nm (Fig. A3, Table A1). This is because these variables are 'determined' as the values saved in the LUT nodes that minimizes the standard deviation of

AOD assuming each aerosol model. More specifically, the ancillary variables are calculated by mixing the top three weighted node values that minimizes the standard deviation. Theoretically, if the TOA reflectance, radiative transfer model, the algorithm, and the LUT perfectly reflects the real world, the determined values would be identical to the true values. However, mainly due to errors in aerosol models and their assumed aerosol properties, the aerosol model that minimizes the AOD of each aerosol model may not always hold aerosol optical properties that the real world exhibits. Here, the FMF of GOCI high-resolution products has underestimation issues over AERONET FMF > 0.6 and performs better at cases where aerosol particles are large (AERONET FMF < 0.4). The underestimation of GOCI FMF has led some fine-mode aerosols of AERONET classified into coarse-mode types of GOCI (Table A1). For SSA, majority of the collocations locate in where both GOCI and AERONET show values between 0.9 and 1.

Nevertheless, ancillary products can be qualitatively used to interpret the aerosol distributions. Figure A4 shows a dust plume covering the Yellow Sea and the Korean Peninsula on May 28th, 2014, which is when yellow dust was identified by the Korea Meteorological Administration. Note that at points where AOD has a negative value, the ancillary variables are represented as NaN. Over the region where the dust plume is located, the aerosol type is mostly classified as dust, and some pixels were identified as mixture (Fig. A4c). Low AE and FMF values (Fig. A4d and A4f) indicate the coarse size of the aerosols included in the dust plume, and low SSA (Fig. A4e) indicate that these aerosols are less absorbing in 440 nm. However, it should be noted that because the ancillary variables are provided from pre-determined nodes, their spatial distributions are somewhat discrete. Overall, ancillary variables of GOCI high-resolution product may be useful for interpreting the relative size and scattering properties used to yield AOD, but the qualitative usage should be taken with care.

## 3.3 Decadal AOD trend analysis

The GOCI high-resolution YAER AOD dataset provides consistent decadal information over East Asia, during a period in which AOD was declining steeply, and also includes diurnal variation of AOD. Figure 8 shows the average GOCI AOD (Fig. 8a) and the trends of AOD (Fig. 8b) for each product resolution pixel. When calculating averages in Fig. 8a, hourly retrievals were averaged into daily average, and the daily averages of 10 years were averaged to yield Fig. 8a. When calculating trends in Fig. 8b, the trend was calculated by linearly interpolating the yearly average AOD from 2012 to 2020 to remove the effects of monthly variations and seasonality. Note that 2012 and 2020 were used to ensure the robustness because for 2011, the observation began in March, and for 2021, the observation ended in March. The average AOD over East Asia revealed regions where AOD increments stand out compared to surrounding regions; vast regions of Eastern China and most populated and industrialized cities of South Korea and Japan had a higher AOD than the background. This clearly implies the non-negligible effect of anthropogenic aerosols over East Asia throughout the decade (Hu et al., 2017). Compared to the near-zero AOD of the Pacific Ocean, AOD over the Yellow Sea (between Eastern China and the Korean Peninsula) showed higher values between 0.4–0.6, which can be attributed to the long-range transport of aerosols of Eastern China along the westerlies (Lee et al., 2019; Lee et al., 2021).

For most pixels, the GOCI high-resolution YAER AOD decreased throughout 2012 to 2020 (Fig. 8b). The largest decrease of AOD was identified over Eastern China, especially in the Beijing, Tianjin, Hebei, and Shandong provinces (hereinafter the BTH region and its surroundings). This result aligns with the result of previous studies using LEO satellites and reanalysis during similar time periods (Sogacheva et al., 2018; Sun et al., 2019; Li, 2020). The ΔAOD year<sup>-1</sup> calculated from earlier studies slightly varies, where our dataset shows about -0.05 ΔAOD year<sup>-1</sup> over the BTH region and its surroundings, whereas
Sogacheva et al. (2018) and Sun et al. (2019) implied approximately -0.07 ΔAOD year<sup>-1</sup> (for 2011–2017) and -0.03 ΔAOD year<sup>-1</sup> (for 2010–2017), respectively (see their Fig. 7 and Fig. 11). The minor discrepancies between the studies are due to the difference in the dataset itself, the spatial and temporal resolution of the dataset, and the period used in the analysis.

Figure 8. (a) Average GOCI high-resolution YAER AOD during the whole observation period and (b) the trends of AOD (ΔAOD year<sup>-1</sup>) for each pixel. For (b), only the pixels with p-values less than 0.05 are shown in colors.

The trend of AOD for each city is shown with its monthly averages (Fig. 9). As time progresses, AOD displayed a decreasing trend in the order of Beijing, Pyongyang, Seoul, and Tokyo. The p-value of each city's trend was 0.2849, 0.1896, 0.0008, 0.0039 for Seoul, Tokyo, Pyongyang, and Beijing, respectively. While this indicates that for Beijing and Pyongyang, the decreasing trend of AOD is statistically significant (p < 0.05) for GOCI observation period, the trends and their statistical significance may vary depending on factors such as the definition of city boundaries and the averaging method. The Pearson correlation between the monthly average of cities was high (R > 0.7) between pairs of Seoul and Pyongyang (0.711) and Tokyo and Beijing (0.724), and low (R 

Figure 9. Trends of GOCI high-resolution YAER AOD for Seoul (red), Tokyo (yellow), Pyongyang (green), and Beijing (blue). The grid boxes of each city are marked in Fig 1.

## 400 4 Data availability

The GOCI high-resolution YAER aerosol optical properties is saved in NetCDF-4 format and is publicly available at https://doi.org/10.5281/zenodo.16656274 (Lee et al., 2025). The hourly NetCDF-4 files can be accessed by opening the monthly zipped file. The hourly data ranges from March 1<sup>st</sup>, 2011, 00:16:42 UTC to March 31<sup>st</sup>, 2021, 07:16:41 UTC. The geospatial variables, namely longitude, latitude, VZA, the viewing azimuth angle, and the land-sea mask at the product resolution are provided as a separate file, named as "GOCI\_YAERAERO\_hires\_navigation.nc". The whole dataset shares the same coordinates and geospatial variables.

#### **5** Conclusion

The GOCI high-resolution aerosol dataset was designed to meet recent user demands and provide enhanced information on aerosol optical properties over long term. The high-resolution dataset encompasses Northeastern Asia, the region where the characteristics and distribution of aerosols were complex during the GOCI observation period of March 2011 to March 2021. The dataset is provided at one-hour intervals, from 00:30 UTC to 07:30 UTC, which is the daytime in the local area. The hourly dataset includes five properties: the AOD, FMF, SSA, AE, and aerosol type. The high-resolution product has 2 km × 2 km spatial resolution at nadir, which is nine times finer than the 6 km resolution of the operational version. The dataset is available online at https://doi.org/10.5281/zenodo.16656274 (Lee et al., 2025) in NetCDF-4 format.

- The GOCI high-resolution YAER algorithm mainly consists of three steps, namely cloud detection and pixel aggregation, classification of surface regimes, and inversion. Hourly radiances observed from eight channels of GOCI are used as a major input for the retrieval. First, multiple criteria including spectral reflectance, its standard deviations, and thresholds are adopted to screen out unsuitable pixels at the observation resolution for aerosol retrieval, such as cloudy, snowy, and bright surface pixels over both land and ocean. Then, the remaining observation resolution pixels are aggregated into the product resolution.
- Each pixel is assigned into one of the three regimes: land, ocean, and turbid water. The darkest 1-3% of RCR is regarded as the surface reflectance for land and turbid water, whereas the climatology of wind speed is used to calculate the surface reflectance for the ocean. Inversion is conducted using a pre-calculated LUT, where the AOD is retrieved as a primary product, and the FMF, SSA, AE, and aerosol type are calculated as by-products.
- The characteristics of GOCI high-resolution dataset were investigated. Over the ocean where clouds leave clear-sky pixels in between, the high-resolution products show more intricate features around clouds. Around the coastlines of the Korean Peninsula where turbid water and red tides frequently occur, the high-resolution products demonstrate a more refined quality compared to the operational version. While discrete discontinuity occurs in the operational version due to a mixed signal of land and ocean being reflected in a single pixel, the high-resolution pixels provide a more realistic view of the coastal AOD. In addition, a high-AOD event over SMA was analyzed at the pixel level. Compared to 6 km pixels which cover one or two sub-districts, the high-resolution pixels capture variations of AOD within individual sub-district in SMA. The high-resolution products excluded the influence of inland water pixels, which can introduce unexpected biases at coarser resolutions.
  - The trends and distributions of decadal high-resolution dataset were analyzed from various perspectives. During the 2010s, some East Asian countries tightened their regulations on air pollutant emissions to improve air quality and promote public health. The impact of these regulations was captured by the GOCI high-resolution dataset; although most of the GOCI FOR experienced a significant decrease in AOD, this trend was most pronounced in Eastern China. When averaged monthly, AOD peaked at summertime, reflecting the effect of hygroscopic growth of aerosols during humid East Asian summer.

- We validated the GOCI high-resolution aerosol products to AERONET, a ground-truth reference dataset, by collocating the two datasets spatially and temporally. The decadal GOCI high-resolution AOD divided into four seasons showed good agreement to AERONET direct sun products, showing the % within EE as high as 64.5%. The bias of GOCI high-resolution AOD against AERONET AOD was analyzed in terms of AERONET AOD, NDVI, scattering angle, and GOCI observation time. Most AOD bias was constrained within the ±0.05 envelope, except for an overestimation issue over turbid water. Most notably, the retrieval algorithm and the resulting product maintained its performance throughout the observation times, proving its diurnal stability.
- The proposed GOCI high-resolution YAER aerosol products are anticipated to be applied in various studies. For example, this data has been utilized in producing a high-resolution PM<sub>2.5</sub> dataset, providing a more detailed view of surface-level air pollution over East Asia (Pendergrass et al., 2025). With detailed information on clear-sky pixels around clouds, the dataset can be incorporated into aerosol-cloud interaction studies (Myhre et al., 2007; Bai et al., 2020). With its advanced quality along coastlines, the high-resolution products can contribute to a more accurate atmospheric correction over these areas and may

assist research on the relationship between AOD and marine microorganisms such as chlorophyll-a (Shen et al., 2020). Furthermore, since the high-resolution dataset has a similar spatial resolution to GOCI-II, an analysis combining GOCI and GOCI-II would reveal an insight into long-term AOD trends and diurnal variations.

Overall, the GOCI high-resolution YAER aerosol products have high spatiotemporal resolution, promising accuracy, and long-term records to reveal valuable information on aerosol characteristics and distribution over East Asia. Further studies using the algorithm may be conducted on recently launched satellites such as GOCI-II to magnify the usage of this state-of-the-art satellite. Improved noise filtering techniques from machine learning or deep learning may allow for retrieval on higher resolutions.

### **Author contribution**

JWL and SL performed dataset production, curation and evaluation. JK managed project administration, supervised the project, acquired funding. SL, MC, and JHL designed the methodology and analyzed the data. MC, JHL, DJ, SK, and YP provided guidance and revised the manuscript. YP provided support as project manager throughout the GOCI mission. All authors contributed to authorship, participated in discussions, and provided advice on the manuscript.

# **Competing interests**

The contact author has declared that none of the authors has any competing interests.

#### Acknowledgements

This material is based on work supported by Samsung Advanced Institute of Technology and National Research Foundation of Korea (NRF) grant funded by the Korea government (MSIT). This work was supported by the Yonsei Fellow Program, funded by Lee Youn Jae. The authors would also like to acknowledge the support by the Korea Institute of Ocean Science and Technology (KIOST) for providing GOCI observations. We thank the PIs and their staff for establishing and maintaining the AERONET sites used for LUT generation and validation of the retrieved properties in this study.

# 470 Financial Support

This work was supported by Samsung Advanced Institute of Technology (2021-11-2232). This work was supported by the National Research Foundation of Korea (NRF) grant funded by the Korea government (MSIT) (RS-2024-00346149).

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

Figure A1. Same as Fig. 5, but for land pixels.

Figure A2. Same as Fig. 5, but for ocean pixels.

690

Table A1. Confusion matrix of GOCI high-resolution aerosol type compared to AERONET during the whole observation period.

|         |              | GOCI high-resolution |           |         |           |             |           |
|---------|--------------|----------------------|-----------|---------|-----------|-------------|-----------|
|         | Aerosol type | Dust                 | Non-      | Mixture | Highly-   | Moderately- | Non-      |
|         |              |                      | absorbing |         | absorbing | absorbing   | absorbing |
|         |              |                      | coarse    |         | fine      | fine        | fine      |
| AERONET | Dust         | 185                  | 11        | 0       | 0         | 0           | 0         |
|         | Non-         | 2                    | 1         | 0       | 0         | 0           | 0         |
|         | absorbing    |                      |           |         |           |             |           |
|         | coarse       |                      |           |         |           |             |           |
|         | Mixture      | 322                  | 34        | 4       | 1         | 0           | 3         |
|         | Highly-      | 284                  | 358       | 138     | 0         | 1           | 97        |
|         | absorbing    |                      |           |         |           |             |           |
|         | fine         |                      |           |         |           |             |           |
|         | Moderately-  | 405                  | 548       | 644     | 0         | 64          | 687       |
|         | absorbing    |                      |           |         |           |             |           |
|         | fine         |                      |           |         |           |             |           |
|         | Non-         | 209                  | 304       | 362     | 2         | 78          | 1174      |
|         | absorbing    |                      |           |         |           |             |           |
|         | fine         |                      |           |         |           |             |           |

Figure A4. (a) GOCI RGB composite image, (b) AOD, (c) aerosol type, (d) AE, (e) SSA, (f) FMF of May 28th, 00 UTC, 2014.

Figure A5 shows a validation of GOCI high-resolution AOD, MODIS MAIAC AOD, and VIIRS Deep Blue AOD to AERONET AOD during 2015. The spatial resolution of MODIS MAIAC and VIIRS Deep Blue AOD are 1 km and 6 km, respectively. The AOD of 04 UTC, which is approximately 01 UTC over Korea and Japan, is also validated for comparison with LEO satellites. The GOCI high-resolution products have a comparable validation metrics compared to MODIS MAIAC and VIIRS Deep Blue, with a slightly higher error. Despite the error, the prevalence of high AOD pixels of GOCI within EE implies that high AOD plumes frequently went undetected in MODIS and VIIRS, where temporal resolutions are lower. Therefore, using GOCI AOD for air quality analysis can more closely reflect the transport and distribution of aerosols over East Asia.

| Figure A5. Validation of GOCI high-resolution AOD, MODIS MAIAC AOD, and VIIRS Deep Blue AOD of 2015 to AERONET AOD. |
|---------------------------------------------------------------------------------------------------------------------|
|                                                                                                                     |
|                                                                                                                     |
|                                                                                                                     |
|                                                                                                                     |
|                                                                                                                     |
|                                                                                                                     |
|                                                                                                                     |
|                                                                                                                     |
|                                                                                                                     |
|                                                                                                                     |
|                                                                                                                     |
|                                                                                                                     |
|                                                                                                                     |