# Peer review of "A decadal, hourly high-resolution satellite dataset of aerosol optical properties over East Asia"

_Earth System Science Data, 2025_

## Author Comment (AC1)

We thank the reviewers for the helpful comments and their time for reviewing the manuscript. The detailed responses to all the referees are provided below in blue font.

Reviewer 1

This study generates a decade-long (2011-2021) high-resolution aerosol dataset for East Asia using GOCI satellite observations and radiative transfer modeling. The dataset provides robust hourly aerosol optical properties, including aerosol optical depth (AOD), fine-mode fraction (FMF), single-scattering albedo (SSA), Ångström exponent (AE), and aerosol classification. These products are particularly valuable for climate and environmental research communities in improving weather forecasting and air pollution monitoring. While the dataset meets ESSD standards for long-term aerosol records, the manuscript requires major revisions to address some comments before publication.

Response: The authors would like to thank the reviewer for their time, comments, and suggestions. The criticisms and comments have helped us improve our paper. We did our best to answer the raised questions and clarify parts of the manuscript that were found to be unclear or ambiguous. Following, the authors prepared the responses, one by one to the addressed comments.

The GOCI satellite's native 500 m spatial resolution provides unique advantages over other geostationary satellites (e.g., Himawari-8/9, MSG-R, GOES-R) for aerosol monitoring at finer scales. However, the decision to upscale to 2 km resolution - while improving retrieval robustness through pixel grouping - potentially diminishes this competitive advantage. The comparative analysis presented in the study actually demonstrates the superior capability of higher-resolution observations, making the 2 km resolution choice appear scientifically questionable.

Response: We understand the questions arising from selecting 2 km as the retrieval resolution instead of the 500 m observation resolution. While products with high resolution provide the users the information on a finer scale, there are tradeoffs caused from the high resolution. The current 2 km resolution was decided considering the balance between the fine-scale information and its product accuracy.

The tradeoffs of AOD retrieval on the observation resolution can be listed as follows.

First, the stability of retrieval is reduced due to reduced signal-to-noise ratio, increased 3-D cloud effects, and vulnerability to radiometric noise. As stated in Section 2.1.1, the brightest 60% and the darkest 20% of the observed pixels are discarded during the aggregation process, to account for artifacts including cloud 3-D effects and cloud shadows. Thus, using pixels that meet the cloud masking criteria directly for retrieval without aggregation can introduce biases in the resulting AOD. Due to these considerations, many widely used algorithms choose to generate aerosol products at spatial resolutions coarser than the native ground pixel resolution,

even when developing a high-resolution counterpart of an existing operational product (Remer et al., 2013; Sayer et al., 2018).

In addition, although higher spatial resolution can theoretically enhance detail, for the primary end-users of this product, the associated increase in retrieval uncertainty stated above often undermines its benefit, offering limited added value in the context of the intended applications. Due to these reasons, we selected 2 km as the optimal retrieval resolution for this product.

The description above has been added to Section 2.1.1.

Nonetheless, we fully acknowledge the unique value of higher spatial resolution. As such, we noted in the conclusion that foreseen algorithmic advancements (e.g., improved noise filtering, incorporation of machine learning (ML) or deep learning (DL) techniques) may allow for reliable operational retrievals at a higher resolution, which can be addressed in a future work (Section 5).

While AOD validation is thoroughly presented, the derived products (FMF, SSA, AE) lack equivalent validation despite their scientific importance. These parameters should receive proper quantitative evaluation given their utility in aerosol characterization.

Response: We agree that the derived products are of good utility in aerosol characterization, and thus the quantitative validation results for the derived products should be included. The validations of FMF at 550 nm, SSA at 440 nm, AE between 440 nm – 870 nm to AERONET inversion V3 level 2 dataset are conducted. FMF at 550 nm of AERONET was calculated by dividing fine-mode AOD by total AOD at 550 nm obtained from quadratic interpolation of spectral AOD. The spatial and temporal collocation criteria are identical to those of AOD at 550 nm and only points with AOD > 0.3 for FMF and AE and AOD > 0.4 for SSA were used to ensure the necessary sensitivity for these variables (Choi et al., 2018).

[Figure]

Figure A3. Validation of GOCI high-resolution YAER AE between 440–870 nm, SSA at 440 nm, FMF at 550 nm. For SSA, the % within ±0.03 and ±0.05 range are shown, and for FMF and AE, the correlation coefficients are shown.

Table A1. Confusion matrix of GOCI high-resolution aerosol type compared to AERONET during the whole observation period.

| | | GOCI high-resolution | | | | | |
|---|---|---|---|---|---|---|---|
| | Aerosol type | Dust | Non-absorbing coarse | Mixture | Highly-absorbing fine | Moderately-absorbing fine | Non-absorbing fine |
| AERONET | Dust | 185 | 11 | 0 | 0 | 0 | 0 |
| | Non-absorbing coarse | 2 | 1 | 0 | 0 | 0 | 0 |
| | Mixture | 322 | 34 | 4 | 1 | 0 | 3 |
| | Highly-absorbing fine | 284 | 358 | 138 | 0 | 1 | 97 |
| | Moderately-absorbing fine | 405 | 548 | 644 | 0 | 64 | 687 |
| | Non-absorbing fine | 209 | 304 | 362 | 2 | 78 | 1174 |

As for the ancillary variables, the validation results are less promising compared to those of AOD at 550 nm. This is because these variables are 'determined' as the values saved in the LUT nodes that minimizes the standard deviation of AOD assuming each aerosol model. More specifically, the ancillary variables are calculated by mixing the top three weighted node values that minimizes the standard deviation. Theoretically, if the TOA reflectance, radiative transfer model, the algorithm, and the LUT perfectly reflects the real world, the determined values would be identical to the true values (which is assumed as the AERONET values here). However, mainly due to errors in aerosol models and their assumed aerosol properties, the aerosol model that minimizes the AOD of each aerosol model may not always hold aerosol optical properties that the real world does. Here, the FMF of GOCI high-resolution products has underestimation issues over AERONET FMF > 0.6 and performs better at cases where aerosol particles are large (AERONET FMF < 0.4). The underestimation of GOCI FMF has led some fine-mode aerosols of AERONET classified into coarse-mode types of GOCI (Table A1). For SSA, majority of the collocations locate in where both GOCI and AERONET show values between 0.9 and 1. The accuracy of ancillary variables can be improved by developing an advanced algorithms focusing on these variables.

Nevertheless, to strengthen the value of the ancillary products, quantitative analysis on a severe dust case is conducted to provide an example for usage in quantitative analysis.

[Figure]

Figure A4. (a) GOCI RGB composite image and high-resolution (b) AOD, (c) aerosol type, (d) AE, (e) SSA, and (f) FMF products of May 28th, 00 UTC, 2014.

Fig. A4 shows a dust plume covering the Yellow Sea and the Korean Peninsula on May 28th,

2014, which is when yellow dust was identified by the Korea Meteorological Administration. Note that at points where AOD has a negative value, the ancillary variables are represented as NaN. Over the region where the dust plume is located, the aerosol type is mostly classified as dust, and some pixels were identified as mixture (Fig. A4c). Low AE and FMF values (Fig. A4d and A4f) indicate the coarse size of the aerosols included in the dust plume, and low SSA (Fig. A4e) indicate that these aerosols are less absorbing in 440 nm. However, it should be noted that because the ancillary variables are provided from pre-determined nodes, their spatial distributions are somewhat discrete. Overall, ancillary variables of GOCI high-resolution product may be useful for interpreting the relative size and scattering properties within the product, but the qualitative usage should be taken with care.

The qualitative validation and the quantitative case study, as well as the descriptions of ancillary variables are added to the manuscript (Section 3.2).

The figures and tables require significant improvement.

Response: The figures and tables were improved according to the comments.

**Specific comments:**

1. Line 95, the advantage of GOCI, including its visible spectrum, long-term observation records with stability, and the presence of a successor satellite, is not fully shown. The developed retrieval algorithm uses only limited bands rather than the full spectrum. Compared to GOCI, the MSG satellite series have longer observation records and multiple successor satellites.

Response: Owing to its multiple visible bands, GOCI can efficiently separate the signal between the atmosphere and the Earth surface. In addition, the long-term records of aerosols can be obtained when a product from a same algorithm of a successor satellite exists. These advantages are more clarified in Section 1, Line 100.

The inversion is divided into three parts: land, ocean, and turbid water. The wavelengths used for each part is decided based on their stability on separating the signals between the surface and aerosol. For land, wavelengths 412 to 680 nm are used, because the two NIR wavelengths have a higher surface reflectance and thus may cause uncertainties in separating the signals between the land surface and atmospheric aerosols. For ocean and turbid water, the green and red channels are excluded due to their high water-leaving radiances. Overall, all eight bands are incorporated in the retrieval algorithm. The description on wavelength selection was added in Section 2.1.3, Line 177.

While other GEO satellites that have a longer observation records and multiple successor satellites exists, GOCI has its unique values on 1. observes East Asia, where aerosol loadings have been a big concern during the past decade, 2. has a reliable accuracy in terms of AOD. The existence of GEO satellites for other regions of the world, and the value of GOCI for

watching East Asia is added in Section 1, Line 73. The existence of a successor satellite, GOCI-II, and the potential value of comprehensive usage of the GOCI series has been described in Section 5.

2. Line 125, how is the cloud threshold determined? Only three tests are used—is this too few? And how is the water detection threshold used? Confirm whether they align with the operational algorithm's implementation.

Response: While the cloud detection schemes can be roughly classified into three, which are TOA reflectance threshold test, reflectance ratio test, and the $3 \times 3$ pixel standard deviation and mean test, there are multiple wavelengths and ratios that are considered. Total of eleven tests are applied for cloud masking over land and ocean combined. The thresholds are determined based on the references of each cloud detection scheme considering its compatibility to GOCI. The cloud detection thresholds and criteria of the high-resolution algorithm aligns with that of the operational counterpart. The details have been added to Section 2.1.1.

3. Lines 140-145, the minimum reflectance technique often leads to underestimation of surface reflectance since no day can truly be aerosol-free, even if it is very clean.

Response: The 5-year period provides plenty of observations to find a clear scene, but it should be noted that no day can truly be aerosol-free, and the least amount of aerosol present in the 'clear scene' may cause a marginal overestimation of surface reflectance. The overestimation of surface reflectance occurs because the minimal aerosol reflectance of a clear day would be attributed to the surface reflectance, leading to a higher RCR compared to the real world. The error source of surface reflectance database was added to Section 2.1.2, Line 156.

4. Line 155, define and explain the look-up table (LUT) methodology for readers who may be unfamiliar with this technique, despite its common use in the field.

Response: The definition and explanation of the LUT methodology was added in Section 2.1.3, Line 170.

5. Figure 4, why are the gaps different between the two?

Response: The pixels excluded from retrieval are different for high-resolution products and operational products because of the aggregation process. The respective 6 km pixels including the Han River were not screened out because although the inland water pixels in the observation resolution were masked out successfully, the adjacent land pixels within the same 6 km pixel met the aggregation criteria. In other words, some pixels that are included in the operational product but not retrieved in the high-resolution product may result from a mixture of valid and invalid observation resolution pixels. These mixed pixels may fail to meet the threshold for

merging in the high-resolution retrieval but are included in the coarser operational resolution, which is merged from a greater number of pixels. As a result, certain pixels in the operational product may reflect signals from only a very small portion within the 6 km area. Therefore, it can be inferred that the 2 km products effectively eliminate the influence of inappropriate pixels in fine geographical features. The description regarding the difference has been supplemented in Section 3.1, Line 260.

6. Line 255, the uncertainty of AERONET AOD should be updated based on the latest studies.

Response: Thank you for pointing this out. According to Sinyuk et al. (2020), the uncertainty of computed AOD, mainly due to calibration uncertainty, is ~0.010 – 0.021 for field instruments (where the high errors are from UV). We updated the description of AOD uncertainties according to this reference.

7. Figure 6, why are only R, EE, and MBE shown?

Response: To strengthen the analysis, three additional parameters, MAE, RMSE, and number of collocated records (N) are added to Fig. 6.

8. Line 320, the time-series AOD changes (Fig. 9) should be explained in more detail, as they appear very interesting, particularly the intra-annual variations.

Response: The details of time-series AOD are analyzed in depth. The decreasing trend and the p-value of the trend of each city is calculated. Furthermore, the correlations between the four cities are analyzed. While there are some intra-annual variations, several factors such as dust outbreaks, wildfire aerosol plume transports, cloud coverage, meteorology and monsoons may have influenced the trends. The enhanced analysis within the scope of data description was added.

9. I encountered difficulties accessing the dataset (https://doi.org/10.7910/DVN/WWLI4W) during my review. Could this be due to regional access restrictions in China, or is there another technical issue with the data repository?

Response: Thank you for pointing out. We tested the access availability of the repository in multiple countries and found out that for some countries, the access is restricted. We re-uploaded the dataset to a different repository and the manuscript was updated accordingly (https://doi.org/10.5281/zenodo.16656274).

**Technical corrections:**

1. Table 1, suggest to improve by maintaining consistent line spacing throughout all paragraphs in the table.

Response: The line spacing of Table 1 is rearranged.

2. Figure 1, the city legend is missing and should be included.

Response: The city legend was updated.

3. Figure 3 requires descriptive titles and proper labels for each subplot.

Response: The UTCs, spatial resolution, and the observation date of each subplot were labeled.

4. Figure 4, a legend identifying the sites should be provided, and the figure appears excessively long and could benefit from resizing or reorganization.

Response: The legend is added, and the figure is resized.

**References**

Choi, M., Kim, J., Lee, J., Kim, M., Park, Y. J., Holben, B., ... & Song, C. H. (2018). GOCI Yonsei aerosol retrieval version 2 products: an improved algorithm and error analysis with uncertainty estimation from 5-year validation over East Asia. *Atmospheric Measurement Techniques*, *11*(1), 385-408.

Remer, L. A., Mattoo, S., Levy, R. C., & Munchak, L. A. (2013). MODIS 3 km aerosol product: algorithm and global perspective. *Atmospheric Measurement Techniques*, *6*(7), 1829-1844.

Sayer, A. M., Hsu, N. C., Lee, J., Bettenhausen, C., Kim, W. V., & Smirnov, A. J. J. O. G. R. A. (2018). Satellite Ocean Aerosol Retrieval (SOAR) algorithm extension to S-NPP VIIRS as part of the "Deep Blue" aerosol project. *Journal of Geophysical Research: Atmospheres*, *123*(1), 380-400.

---

## Author Comment (AC2)

We thank the reviewers for the helpful comments and their time for reviewing the manuscript. The detailed responses to all the referees are provided below in blue font.

Response 2

This manuscript presents a valuable decadal, hourly, high-resolution (2 km) aerosol optical properties (AOPs) dataset for East Asia, derived from GOCI satellite observations and the Yonsei Aerosol Retrieval Algorithm (YAER). The dataset covers 2011–2021 and includes AOD, fine mode fraction (FMF), single scattering albedo (SSA), Ångström exponent (AE), and aerosol type. The paper highlights improvements in spatial/temporal coverage and retrievals over challenging environments (clouds, coastlines) and provides validation against AERONET. Such a dataset will undoubtedly benefit climate, air quality, and health research in the region. The manuscript is generally well-structured, and the methods and results are clearly described. However, several issues require attention before publication.

Response: The authors would like to thank the reviewer for their time, comments, and suggestions. The criticisms and comments have helped us improve our paper. We did our best to answer the raised questions and clarify parts of the manuscript that were found to be unclear or ambiguous. Following, the authors prepared the responses, one by one to the addressed comments.

**Major comments:**

1. The manuscript notes that FMF, SSA, AE, and aerosol type are ancillary and "recommended for qualitative or interpretive use" due to uncertainty. However, the practical scientific utility of the dataset would be significantly enhanced by providing quantitative validation and uncertainty characterization for these variables (not just for AOD). Please include additional validation results for FMF, SSA, AE, and aerosol type where possible (even if only for selected periods/sites with ground truth), and discuss sources of error and their implications for users.

Response: We agree that the derived products are of good utility in aerosol characterization, and thus the quantitative validation results for the derived products should be included. The validations of FMF at 550 nm, SSA at 440 nm, AE between 440 nm – 870 nm to AERONET inversion V3 level 2 dataset are conducted. FMF at 550 nm of AERONET was calculated by dividing fine-mode AOD by total AOD at 550 nm obtained from quadratic interpolation of spectral AOD. The spatial and temporal collocation criteria are identical to those of AOD at 550 nm and only points with AOD > 0.3 for FMF and AE and AOD > 0.4 for SSA were used to ensure the necessary sensitivity for these variables (Choi et al., 2018).

[Figure]

Figure A3. Validation of GOCI high-resolution YAER AE between 440–870 nm, SSA at 440 nm, FMF at 550 nm. For SSA, the % within ±0.03 and ±0.05 range are shown, and for FMF and AE, the correlation coefficients are shown.

Table A1. Confusion matrix of GOCI high-resolution aerosol type compared to AERONET during the whole observation period.

| | | GOCI high-resolution | | | | | |
|---|---|---|---|---|---|---|---|
| | Aerosol type | Dust | Non-absorbing coarse | Mixture | Highly-absorbing fine | Moderately-absorbing fine | Non-absorbing fine |
| AERONET | Dust | 185 | 11 | 0 | 0 | 0 | 0 |
| | Non-absorbing coarse | 2 | 1 | 0 | 0 | 0 | 0 |
| | Mixture | 322 | 34 | 4 | 1 | 0 | 3 |
| | Highly-absorbing fine | 284 | 358 | 138 | 0 | 1 | 97 |
| | Moderately-absorbing fine | 405 | 548 | 644 | 0 | 64 | 687 |
| | Non-absorbing fine | 209 | 304 | 362 | 2 | 78 | 1174 |

As for the ancillary variables, the validation results are less promising compared to those of AOD at 550 nm. This is because these variables are 'determined' as the values saved in the LUT nodes that minimizes the standard deviation of AOD assuming each aerosol model. More specifically, the ancillary variables are calculated by mixing the top three weighted node values

that minimizes the standard deviation. Theoretically, if the TOA reflectance, radiative transfer model, the algorithm, and the LUT perfectly reflects the real world, the determined values would be identical to the true values (which is assumed as the AERONET values here). However, mainly due to errors in aerosol models and their assumed aerosol properties, the aerosol model that minimizes the AOD of each aerosol model may not always hold aerosol optical properties that the real world does. Here, the FMF of GOCI high-resolution products has underestimation issues over AERONET FMF > 0.6 and performs better at cases where aerosol particles are large (AERONET FMF < 0.4). The underestimation of GOCI FMF has led some fine-mode aerosols of AERONET classified into coarse-mode types of GOCI (Table A1). For SSA, majority of the collocations locate in where both GOCI and AERONET show values between 0.9 and 1. The accuracy of ancillary variables can be improved by developing an advanced algorithms focusing on these variables.

Nevertheless, to strengthen the value of the ancillary products, quantitative analysis on a severe dust case is conducted to provide an example for usage in quantitative analysis.

[Figure]

Figure A4. (a) GOCI RGB composite image and high-resolution (b) AOD, (c) aerosol type, (d) AE, (e) SSA, and (f) FMF products of May 28th, 00 UTC, 2014.

Fig. A4 shows a dust plume covering the Yellow Sea and the Korean Peninsula on May 28th,

2014, which is one of yellow dust cases identified by the Korea Meteorological Administration. Note that at points where AOD has a negative value, the ancillary variables are represented as NaN. Over the region where the dust plume is located, the aerosol type is mostly classified as dust, and some pixels were identified as mixture (Fig. A4c). Low AE and FMF values (Fig. A4d and A4f) indicate the coarse size of the aerosols included in the dust plume, and low SSA (Fig. A4e) indicate that these aerosols are less absorbing in 440 nm. However, it should be noted that because the ancillary variables are provided from pre-determined nodes, their spatial distributions are somewhat discrete. Overall, ancillary variables of GOCI high-resolution product may be useful for interpreting the relative size and scattering properties within the product, but the qualitative usage should be taken with care.

The qualitative validation and the quantitative case study, as well as the descriptions of ancillary variables are added to the manuscript (Section 3.2).

2. The authors describe an advanced cloud detection and removal scheme but do not provide a systematic assessment of residual cloud contamination, which is known to bias AOD retrievals, especially at high resolution. It would be valuable to compare the performance of the new cloud screening to that of standard (operational) products.

Response: We acknowledge the reviewer's concern; however, the cloud detection and removal scheme used in the high-resolution algorithm and the operational algorithm is identical, with no modifications. The reason for the pronounced fine-scale AOD between cloud structure is mainly due to the aggregation process where pixels of observation resolution are aggregated into the product resolution. Thus, the AOD details among clouds are not the results of advanced cloud detection but are from the higher product resolution. Residual clouds may be represented as high AOD, which are shown in Fig. 5, where some points have high GOCI AOD at low AERONET AOD.

As an alternative, comparisons with cloud products from other GEO satellites can be attempted; however, the high-resolution aerosol product only provides aerosol information, and pixels that are considered unsuitable for aerosol retrieval has been discarded. These discarded pixels include not only clouds but also contains overly bright surfaces, sun glints, and observation defects. Moreover, temporal differences between different satellites can lead to large discrepancies in cloud positions and distributions. Therefore, a strict apple-to-apple comparison of cloud detection is difficult to achieve.

To clarify, the description on cloud detection algorithm is more specified in Section 2.1.1, and the impact of cloud contamination has been described in Section 3.2.

3. While the validation of AOD against AERONET is comprehensive, the data show persistent underestimation at high AOD (>1.0) and overestimation over turbid water and sparsely vegetated land. The current discussion is brief. Expand the discussion on possible causes for these retrieval biases, such as surface reflectance a priori selection, aerosol model assumptions,

or radiative transfer LUT limitations. Suggest potential mitigation strategies, or at least clarify limitations for high-AOD and complex surface regimes.

Response: Figure 7a, 7b shows a persistent underestimation of high AOD and overestimation over turbid water and sparsely vegetated land. The underestimation of high AOD is primarily attributed to errors in aerosol model selection and the errors of assumed aerosol optical properties within the LUT. For example, because the non-absorbing coarse model is assumed from higher SSA compared to highly-absorbing fine type, if a highly-absorbing fine aerosol is mistaken as non-absorbing coarse type particle, the retrieved AOD would be lower. When validating aerosol type, some portion of highly-absorbing fine aerosols, which mostly arise from smoke and exhibits high AOD, were being mistaken as dust and non-absorbing coarse type (Table A1), which would have led to AOD underestimation.

Overestimation of AOD over turbid water and sparsely vegetated land is mainly due to the errors of surface reflectance a priori selection. Both surfaces are bright over visible spectra. This indicates that for these bright surfaces, the minimum reflectance technique has limitations, and vegetation should be considered in calculating the surface reflectance.

The description on the limitations and possible causes for the biases has been expanded (Section 3.2, Line 316).

4. While the manuscript references related studies and datasets (e.g., MODIS, VIIRS, GEMS), a direct comparison with other existing satellite AOD products over the same region and period (where possible) would further contextualize the strengths and weaknesses of the new GOCI dataset. It should provide a comparative assessment (statistical metrics, spatial patterns, or time series) between your dataset and other satellite or reanalysis products (e.g., MODIS MAIAC, VIIRS DB, GEMS) to demonstrate added value and highlight specific improvements or tradeoffs.

Response: Figure A5 shows a validation of GOCI high-resolution AOD, MODIS MAIAC AOD, and VIIRS Deep Blue AOD to AERONET AOD during 2015. The spatial resolution of MODIS MAIAC and VIIRS Deep Blue AOD are 1 km and 6 km, respectively. The AOD of 04 UTC, which is approximately 01 UTC over Korea and Japan, is also validated for comparison with LEO satellites. GEMS was not included in the analysis because the observation period of GEMS (November 2020–) and GOCI (–March 2021) overlap for only a short time.

The GOCI high-resolution products have a comparable validation metrics compared to MODIS MAIAC and VIIRS Deep Blue, with a slightly higher error. Despite the error, the prevalence of high AOD pixels of GOCI within EE implies that high AOD plumes frequently went undetected in MODIS and VIIRS, where temporal resolutions are lower. Therefore, using GOCI AOD for air quality analysis can more closely reflect the transport and distribution of aerosols over East Asia.

The description and Fig. A5 has been added to Appendix A.

[Figure]

**Figure A5. Validation of GOCI high-resolution AOD, MODIS MAIAC AOD, and VIIRS Deep Blue AOD of 2015 to AERONET AOD.**

**Minor comments:**

1. Please clarify how gaps due to clouds, sun-glint, or other data removal are handled (e.g., are missing values flagged, interpolated, or left as NaN?).

Response: The removed pixels are left as NaN. The description has been added in Section 2.2, Line 201.

2. Improve the clarity of several figures (e.g., Figures 2–4): add scale bars, colorbars, and clearer labeling for readers less familiar with the region.

Response: Figures 2–4 are revised to have a clearer labelling and information.

**References**

Choi, M., Kim, J., Lee, J., Kim, M., Park, Y. J., Holben, B., ... & Song, C. H. (2018). GOCI Yonsei aerosol retrieval version 2 products: an improved algorithm and error analysis with uncertainty estimation from 5-year validation over East Asia. *Atmospheric Measurement Techniques*, *11*(1), 385-408.